# CONSISTENCY CALIBRATION:
# IMPROVING UNCERTAINTY CALIBRATION VIA CONSISTENCY AMONG PERTURBED NEIGHBORS

## ABSTRACT

Calibration is crucial in deep learning applications, especially in fields like healthcare and autonomous driving, where accurate confidence estimates are vital for decision-making. However, deep neural networks often suffer from miscalibration, with reliability diagrams and Expected Calibration Error (ECE) being the only standard perspective for evaluating calibration performance. In this paper, we introduce the concept of *consistency* as an alternative perspective on model calibration, inspired by uncertainty estimation literature in large language models (LLMs). We highlight its advantages over the traditional reliability-based view. Building on this concept, we propose a post-hoc calibration method called Consistency Calibration (CC), which adjusts confidence based on the model's consistency across perturbed inputs. CC is particularly effective in locally uncertainty estimation, as it requires no additional data samples or label information, instead generating input perturbations directly from the source data. Moreover, we show that performing perturbations at the logit level significantly improves computational efficiency. We validate the effectiveness of CC through extensive comparisons with various post-hoc and training-time calibration methods, demonstrating state-of-the-art performance on standard datasets such as CIFAR-10, CIFAR-100, and ImageNet, as well as on long-tailed datasets like ImageNet-LT. Code is available at https://anonymous.4open.science/r/Consistency-Calibration-E248.

## 1 INTRODUCTION

Calibration is essential in many deep learning applications where accurate confidence estimates are as important as the predictions themselves. In fields like healthcare Chen et al. (2018) and autonomous driving Feng et al. (2019), decisions often rely not only on the model's output but also on how confident the model is in its predictions. A well-calibrated model should reflect the ground truth uncertainty. In healthcare, for instance, a model that accurately reflects uncertainty can help doctors trust the system's confidence when diagnosing critical conditions.

However, current deep learning models are often found to be miscalibrated (Guo et al., 2017). To evaluate calibration performance, Naeini et al. (2015) introduced ECE, which has become the gold standard, based on the reliability diagram (DeGroot & Fienberg, 1983). Although several improved metrics have since been proposed, such as AdaptiveECE (AdaECE) (Nixon et al., 2019) and Class-wiseECE (CECE) (Kull et al., 2019), they all adopt the same fundamental perspective on calibration: if a model assigns 80% confidence to its predictions, then, ideally, 80% of those predictions should be correct. We refer to this classical approach as the *reliability view*, which seeks to align predicted confidence levels with actual model accuracy.

The concept of *consistency* has gained increasing importance in black-box uncertainty estimation, particularly in recent developments in large language models (LLMs) (Wang et al., 2022; Tam et al., 2022; Xiong et al., 2023b; Geng et al., 2023). If an LLM is confident in its answer, it should provide consistent responses to similar questions. For instance, if an LLM confidently answers the question "What is the answer to 5 + 3?" with "8", it should also consistently provide "8" for the similar question "What is the result of five plus three?" In this paper, we extend this concept of consistency to model calibration, proposing a new perspective of calibration called consistency.

Specifically, in a classification task, if a model is confident in its prediction, it should consistently provide the same output across multiple perturbed versions of the input. Consistency measures how often a model's prediction remains unchanged when the input is perturbed within a small neighborhood. A high consistency score implies that the model's predictions are stable and confident. In this view, a perfectly calibrated model should have its predicted confidence levels align with the consistency observed across these perturbed inputs.

In the following sections, we discuss the differences between calibration from the perspectives of reliability and consistency in Sections 2.1 and 2.2. Section 2.3 highlights the advantages of the consistency approach over the reliability view through a toy example. In Section 2.4, CC is introduced, which involves perturbing the logits. We provide empirical evidence to explain its effectiveness in Section 2.6. Finally, in Section 2.5, we demonstrate that consistency can serve as a reliable method for local uncertainty estimation.

Our contributions can be summarized as follows:

- We introduce a novel perspective on calibration based on consistency and highlight its advantages over traditional reliability view represented by ECE.

- We propose an easy-to-implement and computationally efficient post-hoc calibration method called *Consistency Calibration*, which replaces the original confidence score with a consistency measure calculated from perturbed logits using data neighbors.

- *CC* serves as a reliable and effective method for local uncertainty estimation, as it does not require additional data samples or label information. Instead, it generates data neighborhoods based on the source data.

- We conduct comparisons with multiple post-hoc and training-time calibration methods, demonstrating state-of-the-art performance on standard datasets, including CIFAR-10, CIFAR-100, and ImageNet, as well as in long-tailed scenarios like ImageNet-LT.

## 2 METHODOLOGY

In a classification task, let $\mathcal{X}$ represent the input space and $\mathcal{Y}$ the label space. The neural network $f(\cdot)$ and projection head $g(\cdot)$ maps $x \in \mathcal{X}$ to a vector of logits $z = g(f(x)) \in \mathbb{R}^K$, where each $z_k$ is the logit for class $k$. These logits are then transformed into a probability distribution $\hat{p} = \text{softmax}(z)$ over $K$ classes using the softmax function:

$$\hat{p}_k = \frac{e^{z_k}}{\sum_{i=1}^{K} e^{z_i}}, \quad k = 1, \ldots, K, \tag{1}$$

where $\mathbf{k} = \arg\max_i \hat{p}_i$ denotes the predicted label index. The ground-truth label $y \in \mathcal{Y}$ represents the true class, and $\hat{y} \in \mathcal{Y}$ is the predicted label. The confidence score $\hat{p}_{\mathbf{k}}$ represents the predicted probability assigned to the predicted label $\mathbf{k}$.

### 2.1 CALIBRATION IN THE VIEW OF RELIABILITY

Calibration in the view of reliability has been widely accepted since the introduction of the reliability diagram by DeGroot & Fienberg (1983). In this view, a classifier is considered perfectly calibrated if its predicted confidence $\hat{p}$ accurately represents the true probability of correctness. Formally, this is expressed as:

$$\mathbb{P}(\hat{y} = y \mid \hat{p} = p) = p \quad \text{for all } p \in [0, 1]. \tag{2}$$

In other words, if a model assigns a confidence score of 80%, the prediction $\hat{y}$ should be correct 80% of the time. To move beyond visual inspection of reliability diagram, Naeini et al. (2015) developed a quantitative metric from the reliability diagram called the *Expected Calibration Error* (ECE). ECE provides a more precise measurement of miscalibration by calculating the average discrepancy between a model's predicted confidence and the actual accuracy of predictions at the same confidence level. ECE is defined as:

$$\text{ECE} = \mathbb{E}_{\hat{p}} \left[ |\mathbb{P}(\hat{y} = y \mid \hat{p}) - \hat{p}| \right]. \tag{3}$$

In practice, due to finite sample sizes, an approximation is used by binning predictions into $M$ equally spaced confidence intervals, $\{B_m\}_{m=1}^{M}$. Each bin $B_m$ contains predictions with confidence scores $\hat{p} \in \left[\frac{m}{M}, \frac{m+1}{M}\right)$. For each bin, the average confidence $C_m$ and accuracy $A_m$ are computed as:

$$C_m = \frac{1}{|B_m|} \sum_{i \in B_m} \hat{p}_i, \quad A_m = \frac{1}{|B_m|} \sum_{i \in B_m} \mathbb{1}(\hat{y}_i = y_i), \tag{4}$$

where $\mathbb{1}$ is the indicator function, and $|B_m|$ is the number of samples in bin $B_m$. The approximate ECE is then computed as the weighted average of the absolute difference between bin accuracy and bin confidence:

$$\text{ECE} = \sum_{m=1}^{M} \frac{|B_m|}{N} \left| A_m - C_m \right|, \tag{5}$$

where $N$ is the total number of samples. Several variants of ECE exist. For instance, *AdaECE* uses adaptive binning to ensure equal sample sizes in each bin and avoid the issue of uneven confidence distribution in ECE, while *CECE* computes ECE on a per-class basis, enabling better detection of class-specific calibration errors.

## 2.2 CALIBRATION IN THE VIEW OF CONSISTENCY

We offer an alternative perspective on calibration by examining it through the concept of consistency. In a real-world scenario, an individual confident in their answer tends to maintain that answer, even when faced with external doubts or minor alterations to the question. On the other hand, someone who is uncertain might change their response when presented with slightly misleading information or variations in the question. We define this adherence to the original answer as *consistency*.

Recent advances in LLMs, particularly black-box models utilize *factual consistency* to enhance performance (Wang et al., 2022; Tam et al., 2022; Xiong et al., 2023b; Geng et al., 2023). These studies frame the consistency of a model's responses as an indicator of its uncertainty. In the context of classification tasks, calibration can also be described in terms of consistency. Specifically, for classification models, we can formalize this relationship as follows:

**Proposition 1.** *If a model is confident in its prediction, it should consistently output the same prediction when the input is slightly perturbed. The consistency $c$ of a sample $x$ is defined as*

$$c_k(x) = \frac{1}{T} \sum_{t=1}^{T} \mathbb{1}(\hat{y}(\tilde{x}_t) = k), \text{ where } d(\tilde{x}_t, x) < \epsilon^*, \quad \text{for } k = 1, \ldots, K \tag{6}$$

*where $T$ is the number of perturbed neighbors, $\hat{y}(\tilde{x}_t)$ is the predicted label for the perturbed input $\tilde{x}_t$, and the distance between the original sample $x$ and its perturbed version $\tilde{x}_t$ is smaller than a constant $\epsilon^*$, according to some distance metric $d$. A model is said to be perfectly calibrated if, for all samples $x$, given a suitable set of perturbed neighbors $\{\tilde{x}_t \mid t = 1, \ldots, T\}$, the predicted confidence score $\hat{p}(x)$ satisfies:*

$$\hat{p}_k(x) = c_k(x), \quad \text{for } k = 1, \ldots, K \tag{7}$$

However, identifying a suitable perturbed neighborhood is non-trivial—it is challenging to determine an appropriate constant $\epsilon^*$ and distance metric $d$. Fortunately, in image classification tasks, a perturbed neighbor is often considered a data-augmented version of the original image. Thus, we begin our exploration by using image data augmentation.

To evaluate the effectiveness of consistency-based confidence, we design an experimental setting using a ResNet-50 model trained on CIFAR-10 with data augmentation (RandomCrop and RandomHorizontalFlip). We generate perturbed neighbors by applying various levels of data augmentation to the entire CIFAR-10 test set, creating 100 perturbed neighbors for each test sample. The calibration performance of consistency is assessed on the test set in the following settings:

- *Baseline*: Confidence score is extracted on the original test set, serving as the baseline.
- *Weak Augmentation (Train Augmentation)*: The confidence score is replaced with consistency derived from perturbed neighbors generated using train-time augmentation (RandomCrop and RandomHorizontalFlip), denoted by the yellow star.

- *Moderate Augmentation (Train Augmentation + ColorJitter)*: The confidence score is replaced with consistency measured from perturbed neighbors generated using train-time augmentation and varying strengths of ColorJitter, as indicated by the x-axis values.

- *Stronger Augmentation (Train Augmentation + ColorJitter + Blur)*: The confidence score is replaced with consistency measured from perturbed neighbors generated using train-time augmentation, ColorJitter, and Blur, represented by the red triangle.

The evaluation results are shown in Figure 1a. Consistency using neighbors generated with weak augmentation significantly reduces calibration error compared to the baseline. As we increase the perturbation strength with moderate augmentation, as shown by the x-axis values, the calibration error continues to decrease with minimal impact on accuracy, outperforming the commonly used calibration method, Temperature Scaling, up to a certain perturbation threshold.

However, when moderate augmentation with strength exceeds 0.1, accuracy begins to decline, and ECE increases sharply. With neighbors generated from Stronger Augmentation, both calibration and prediction accuracy deteriorate. This likely occurs because stronger perturbations distort the input to the extent that the model can no longer recognize the data, leading to degraded performance. This suggests that consistency has the potential to provide accurate uncertainty estimates when a suitable perturbed neighborhood is identified.

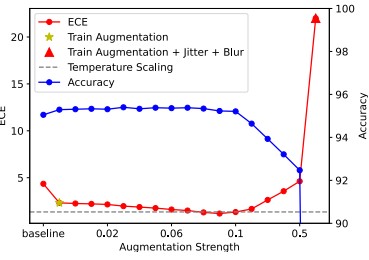
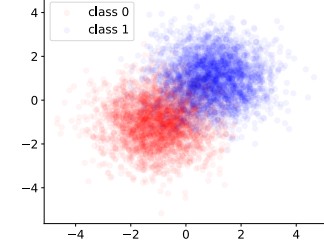
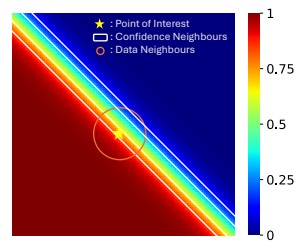

(a) Perturbation applied to images with different augmentations. Consistency calculated using weak or moderate augmentation neighbors significantly reduces calibration error.

(b) Toy dataset generated from two two-dimensional Gaussian distributions. Samples near the diagonal are uncertain to belong to class 0 or class 1.

(c) Heatmap of ground truth uncertainty calculated from the PDF, as given by Eq. 8. Circle and box indicate different neighbourhood selection criteria.

Figure 1: Illustrations of Consistency, Toy Data Distributions, and Ground Truth Uncertainty.

## 2.3 Consistency as a Representation of Ground Truth Uncertainty

On one hand, the *reliability approach* estimates calibration error by comparing the prediction confidence with the average correctness of samples that have similar confidence levels. In this view, the average correctness of such sample neighborhood is treated as an approximation of the ground truth uncertainty. On the other hand, the *consistency approach* directly uses consistency as a measure of ground truth uncertainty. Thus, we are interested in determining which of these two approaches more accurately approximates this uncertainty.

To explore this, we constructed a toy dataset consisting of two two-dimensional Gaussian distributions representing two groups of data: $\mathcal{N}(\mu_0, \Sigma), \mathcal{N}(\mu_1, \Sigma)$ where $\mu_0$ and $\mu_1$ are the mean vectors, and $\Sigma$ is the shared covariance matrix for both groups, labeled 0 and 1, respectively. We generated 1,000,000 data points from each group to form the training dataset, which was used to train a CNN model. An additional 50,000 samples from each group were used to create the test dataset. The input space is $\mathcal{X} = \mathbb{R}^2$, and the label space is $\mathcal{Y} = \{0, 1\}$, as illustrated in Figure 1b.

The ground truth uncertainty, $\eta(x)$, is calculated from the probability density function (PDF) of each distribution:

$$\eta(x) = \frac{p^0(x)}{p^0(x) + p^1(x)} \tag{8}$$

where $p^0(\cdot)$ and $p^1(\cdot)$ are the PDFs of the two distributions. The ground truth uncertainty is illustrated in Figure 1c. For each label, the ground truth confidence can be expressed as $(\eta(x), 1 - \eta(x))$.

In Figure 1c, for a point of interest (marked by a star), the reliability-based approach estimates ground truth uncertainty by calculating the average correctness $A = \frac{1}{|B|} \sum_{i \in B} \mathbb{1}(\hat{y}_i = y_i)$, over a "confidence neighborhood" $B$ (i.e., samples with similar confidence, enclosed by the white boxes), similar to the definition Eq. 4 in ECE. In contrast, the consistency approach estimates uncertainty by considering "data neighborhood," as illustrated by the orange circle. While the reliability approach relies on the availability of multiple data samples within the confidence neighborhood, the consistency approach generates data neighborhoods by perturbing the data.

The key differences between the reliability and consistency views lie in their neighborhood selection criteria $S$ and aggregation methods. The reliability view selects a neighborhood $B$ based on confidence similarity and aggregates the correctness of the samples, while the consistency view selects a neighborhood based on data perturbations and computes consistency, as described in Eq. 9. To compare the two approaches, we evaluate them under three neighborhood selection criteria:

- *Figure 2a*: Reliability view (ECE): $B = \{\tilde{x} \mid |\hat{p}(x) - \hat{p}(\tilde{x})| < \epsilon\}$

- *Figure 2b*: Reliability view (AdaECE): $B = \{\tilde{x} \mid \text{Top K closest confidence neighbors}\}$

- *Figure 2c*: Consistency view (perturbing data): $B = \{\tilde{x} \mid \tilde{x} = x + \epsilon\}$

In Figures 2a and 2b, we use the reliability approach to approximate ground truth uncertainty based on two confidence-neighbor selection criteria. In Figure 2a, we replicate the standard ECE (Guo et al., 2017) approach by selecting confidence neighbors solely based on confidence differences. The x-axis represents the allowed confidence difference between neighbors and the point of interest, while the y-axis shows the average error between estimated and ground truth uncertainty across the test set. In Figure 2b, we replicate the AdaECE (Nixon et al., 2019) approach by selecting the top-K nearest confidence neighbors to estimate uncertainty, with the lowest error (0.57%) achieved by selecting the top 9 nearest neighbors.

In Figure 2c, we apply Gaussian noise $\epsilon$ to perturb the data samples and compute consistency across 100 generated neighbors, with the x-axis representing the noise strength. We compare the uncertainty estimates from the consistency approach with those from the reliability approach. The dashed lines indicate the minimal error achieved by each method. Within a certain range of perturbation strengths, the consistency approach outperforms, yielding a ground truth uncertainty estimation with an overall error as low as 0.3%.

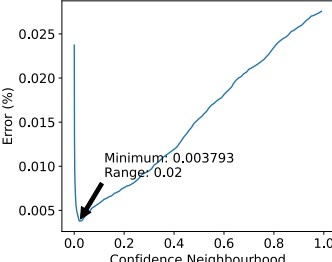 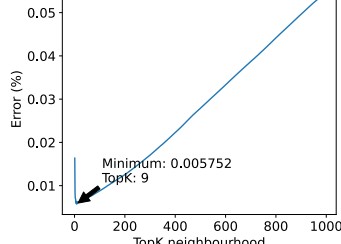 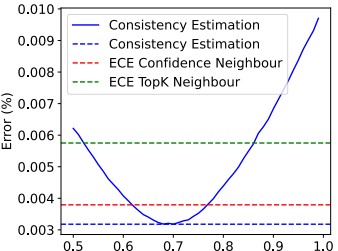

(a) **Reliability view:** Estimating ground truth uncertainty using neighbors with confidence differences indicated on the x-axis.

(b) **Reliability view:** Estimating ground truth uncertainty using the top-k nearest confidence neighbors as indicated on the x-axis.

(c) **Consistency view:** Estimating ground truth uncertainty using data neighbors perturbed within $\epsilon$ as indicated on the x-axis.

Figure 2: Comparison of Consistency vs. Reliability in Estimating Ground Truth Uncertainty

It is important to note that the reliability approach using confidence neighborhoods is essentially equivalent to the ECE measurement, where the allowed confidence gap functions similarly to the hyperparameter "number of bins" in ECE. As shown in Figure 2, the estimation error is sensitive to the allowed confidence gap—meaning that the choice of "number of bins" can significantly impact the ability of ECE to estimate the ground truth uncertainty. Similarly, this sensitivity is also observed in the consistency method, where the strength of perturbation noise affects the uncertainty approximation. Despite this sensitivity, the consistency approach achieves a lower overall estimation error, suggesting its potential as a robust alternative calibration metric.

### 2.4 MORE EFFICIENT CONSISTENCY CALIBRATION

Due to numerous types of data augmentations, determining the optimal perturbation strength using a continuous variable is challenging. To address this, we extend the perturbation process to the feature and logit levels by introducing noise with varying intensities. This approach yields effects similar to those observed with image-level perturbations, as demonstrated in Figure 3a and Figure 3b.

Interestingly, feature- and logit-level perturbations maintain significant calibration performance while offer huge computational advantages. With image-level perturbations, inference must be performed on the entire model $T$ times. In contrast, feature-level require evaluating only the classification head, while logit-level only compute the argmax operation $T$ times. This results in substantial reductions in computational costs. Experiments on other layers can be found in Appendix B.

**Proposition 2.** *We propose a unified definition of our calibration methods, termed Consistency Calibration (CC), which identifies perturbed neighbors at different levels. The calibrated prediction confidence score $\hat{p}'$ is formally defined as:*

$$\hat{p}'_k = \frac{1}{T}\sum_{t=1}^{T}\mathbb{1}\left(\arg\max q\left(\widetilde{h(x)}^t\right) = k\right), \quad for\ k = 1,\dots,K, \tag{9}$$

*where $h(x)$ is the representation of data $x$, $\widetilde{h(x)}^t$ is the perturbed representation, and $q$ is the pipeline to extract the logits $z$.*

Specifically, for data-level perturbations: $h(\cdot) = I(\cdot)$, $\widetilde{h(x)}$ is the augmented data, $q = g(f(\cdot))$. For feature-level perturbations: $h(\cdot) = f(\cdot)$, $\widetilde{h(x)}^t = h(x) + \epsilon_t$, $q = g(\cdot)$. For logit-level perturbations: $h(\cdot) = g(f(\cdot))$, $\widetilde{h(x)}^t = h(x) + \epsilon_t$, $q = I(\cdot)$. Here, $I(\cdot)$ is the identity function, $\epsilon_t$ represents the noise added to features or logits, with its strength determined by minimizing the ECE on a validation set. Given the strong calibration performance and computational efficiency of logit-level perturbations, we refer to logit-level consistency calibration as CC when no specification is provided.

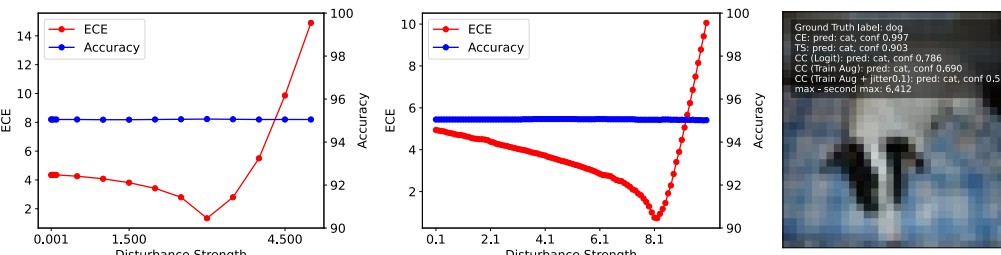

(a) Performance of consistency calibration using data neighbors with feature perturbations at varying noise levels.

(b) Performance of consistency calibration using data neighbors with logit perturbations at varying noise levels.

(c) Performance of local uncertainty estimation using consistency calibration.

Figure 3: Evaluation of Consistency Calibration under Different Perturbation Settings.

### 2.5 CONSISTENCY AS A LOCAL UNCERTAINTY ESTIMATION

Consistency-based methods do not rely on label information or additional data, as they generate their own neighborhood by perturbing the input data. This property allows consistency to serve as a criterion for instance-level uncertainty measurement. As illustrated in Figure 3c, we examine a miscalibrated (incorrect prediction with high confidence) CIFAR-10 test sample, where a ResNet-50 model trained with Cross-Entropy (CE) shows overconfidence, assigning a confidence score of 0.997 despite being incorrect. Using optimal temperature, determined via a validation set, the confidence after temperature scaling decreases slightly, but the model remains overconfident at 0.903.

For comparison, we apply CC by perturbing the logits ("CC (logits)"), applying train time data augmentation ("CC (Train Aug)"), and using a moderate augmentation method ("CC (Train Aug + Jitter)"). The confidence significantly decreases with these approaches. However, too strong augmentations may negatively impact model accuracy, which requires the need for a validation set

to tune the augmentation strength, so we recommend using training-time augmentation to avoid the use of validation set while keeping the prediction accuracy.

Unlike many post-hoc calibration methods that require a large validation set to fine-tune hyperparameters, consistency-based confidence with train-time augmentation can directly provide calibrated confidence scores while maintaining recognizable by models. This approach is particularly valuable in data-limited scenarios, allowing consistency to produce an accurate local uncertainty estimation.

### 2.6 WHY CONSISTENCY CALIBRATION WORKS?

Perturbing images results in straightforward and intuitive image neighborhoods, but the effectiveness of perturbations at the logit level requires further explanation. To understand why logit perturbations work, we examined the differences between highly confident correct predictions and overconfident incorrect ones. These represent well-calibrated and poorly calibrated samples, respectively. During logit disturbance, the label with second-largest logit most likely to become the prediction label. To investigate this, we plotted box plots for both the maximum and second-largest logits for correct and incorrect predictions, as shown in Figure 4a.

For CIFAR-10 test samples, we selected predictions with confidence higher than 99%. We refer to the maximum logit of correct predictions as "Corr. Max" and that of incorrect predictions as "Incorr. Max." Similarly, "Corr. 2nd" represents the second-largest logit of correct predictions, while "Incorr. 2nd" refers to the second-largest logit of incorrect predictions. As shown in Figure 4a, the maximum logit for correct predictions is significantly higher than for incorrect predictions. Additionally, the second-largest logit in correct predictions is much lower than that in incorrect predictions. This indicates that the gap between the maximum and second-largest logits is much larger for correct predictions than for incorrect ones. Despite large difference, due to softmax saturation, the model assigns abnormally high confidence (greater than 99%) to both correct and incorrect predictions, leading to overconfident miscalibration.

Interestingly, we can leverage this difference in the logit gaps between correct and incorrect predictions. Perturbations can easily alter the predictions of overconfident, miscalibrated samples, while having minimal effect on well-calibrated, correct predictions. This different response to perturbations explains why consistency calibration is effective at the logit level. We observed similar patterns in experiments with CIFAR-100 and ImageNet, as shown in Figure 4b and Figure 4c.

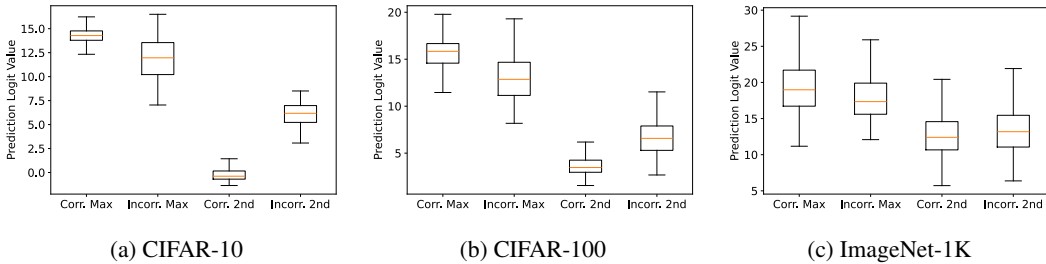

|               |               |                 |
| :-----------: | :-----------: | :-------------: |
| (a) CIFAR-10  | (b) CIFAR-100 | (c) ImageNet-1K |

Figure 4: Distribution of the max logit and second-largest logit for correct and incorrect predictions with more than 99% confidence, representing well-calibrated and miscalibrated samples on ResNet-50 across different datasets. The difference between the max logit and second-largest logit is significantly smaller for miscalibrated samples compared to well-calibrated samples.

## 3 EXPERIMENTS

### 3.1 EXPERIMENTAL SETUP

**Datasets** We conduct experiments on several benchmark datasets, including CIFAR-10, CIFAR-100 (Krizhevsky et al., 2009), and ImageNet (Deng et al., 2009). To assess calibration performance in data-imbalance scenarios, we also include ImageNet-LT (Liu et al., 2019), characterized by its long-tailed class distribution. CIFAR-10 and CIFAR-100 contain 60,000 images of size $32 \times 32$ pixels, with 10 and 100 classes, respectively, split into 45,000 training, 5,000 for validation and 10,000 test images. For ImageNet-1K, we split 20% of the original validation set as the new validation set,

with the remainder used as the test set. We use the $\epsilon$ searched on ImageNet-1k validation set to calibrate ImageNet-LT test set. The testing batch size for all datasets is set to 128.

**Models** We evaluate our approach across various neural network architectures, including ResNet-50 and ResNet-110 (He et al., 2016), Wide ResNet (Zagoruyko & Komodakis, 2016), DenseNet-121 (Huang et al., 2017), and Vision Transformers (ViT-B/16 and ViT-B/32) (Dosovitskiy et al., 2021). These models represent a diverse range of architectures and complexities, allowing us to assess the robustness of our method in different settings. For CIFAR-10 and CIFAR-100, we use pretrained weights from prior work (Mukhoti et al., 2020). All models are trained using stochastic gradient descent (SGD) with a momentum of 0.9 and weight decay of $5 \times 10^{-4}$ for 350 epochs. The learning rate is initialized at 0.1 for the first 150 epochs, reduced to 0.01 for the next 100, and further decreased to 0.001 for the final 100 epochs. For ImageNet, we use pretrained models from PyTorch (Paszke et al., 2019), following the training recipe available on PyTorch's model page.

**Evaluation Metrics and Other Settings** Calibration performance is primarily evaluated using ECE, with additional metrics including AdaECE, CECE, Negative Log-Likelihood (NLL), and top-1 accuracy. All experiments are conducted on an NVIDIA 4090 GPU, with results averaged over five runs to ensure fairness. For all experiments, we set the number of perturbations to $T = 1000$ and search the perturbation strength $\epsilon$ and noise type by minimizing ECE on the validation set.

| Dataset | Model | Vanilla | TS | ETS | PTS | CTS | GC | CC (ours) |
|---------|-------|---------|-----|-----|-----|-----|-----|-----------|
| CIFAR-10 | ResNet-50 | 4.34 | 1.38 | 1.37 | 1.36 | 1.46 | 1.04 | **0.78** |
| | Wide-ResNet | 3.24 | 0.93 | 0.93 | 0.93 | 0.93 | 1.33 | **0.36** |
| CIFAR-100 | ResNet-50 | 17.52 | 5.71 | 5.68 | 5.64 | 6.05 | 3.55 | **1.25** |
| | Wide-ResNet | 15.34 | 4.63 | 4.58 | 4.52 | 4.86 | 2.14 | **1.61** |
| ImageNet-1K | ResNet-50 | 3.76 | 2.09 | 2.09 | 2.08 | 3.14 | 2.54 | **1.53** |
| | DenseNet-121 | 6.59 | 1.64 | 1.66 | 1.68 | 1.94 | 2.51 | **1.48** |
| | Wide-ResNet-50 | 5.49 | 3.03 | 3.04 | 3.04 | 4.13 | 2.16 | **1.33** |
| | Swin-B | 5.02 | 3.90 | 3.90 | 3.93 | 5.43 | 1.61 | **1.58** |
| | ViT-B-16 | 5.61 | 3.61 | 3.62 | 3.64 | 5.50 | 1.75 | **1.66** |
| | ViT-B-32 | 6.40 | 3.76 | 3.78 | 3.84 | 5.74 | **1.39** | 1.72 |
| ImageNet-LT | ResNet-50 | 3.67 | 2.00 | 1.99 | 2.00 | 2.21 | 1.4 | **1.24** |
| | DenseNet-121 | 6.65 | 1.65 | 1.64 | 1.66 | 1.59 | 1.81 | **1.23** |
| | Wide-ResNet-50 | 5.39 | 2.97 | 2.96 | 2.96 | 3.52 | 1.49 | **1.27** |
| | Swin-B | 4.66 | 4.02 | 4.03 | 4.08 | 5.02 | 1.66 | **1.44** |
| | ViT-B-16 | 5.57 | 3.61 | 3.62 | 3.64 | 4.94 | 1.76 | **1.61** |
| | ViT-B-32 | 5.15 | 5.67 | 5.67 | 5.68 | 5.71 | **1.48** | 1.74 |

Table 1: **Comparison of Post-Hoc Calibration Methods Using ECE↓ Across Various Datasets and Models.** ECE values are reported with 15 bins. The best-performing method for each dataset-model combination is in bold, and our method (CC) is highlighted. Results are averaged over 5 runs.

## 3.2 COMPARISON WITH POST-HOC CALIBRATION METHODS

We compare our proposed CC with widely used post-hoc calibration techniques, including Temperature Scaling (TS) (Guo et al., 2017), Ensemble Temperature Scaling (ETS) (Zhang et al., 2020), Parameterized Temperature Scaling (PTS) (Tomani et al., 2022), Class-based Temperature Scaling (CTS) (Frenkel et al., 2021), and Group Calibration (GC) (Yang et al., 2024), as well as uncalibrated models (Vanilla). Our evaluation covers CIFAR-10, CIFAR-100, ImageNet-1K, and ImageNet-LT, using various CNNs and transformers.

**Calibration on Standard Datasets** CC consistently outperforms these methods across CIFAR-10, CIFAR-100, and ImageNet-1K, significantly reducing calibration error. The most notable improvement is seen in CIFAR-100, where CC excels while GC, despite its strong performance on other datasets, struggles. This highlights CC's robustness across datasets with varying complexities. CNNs, which often suffer from overconfidence, are generally well-calibrated with TS-based methods. However, transformers see limited calibration improvements from TS-based methods, with CC outperforming them by a large margin. On larger datasets like ImageNet-1K, CC maintains its advantage. Although GC slightly outperforms CC on ViT-B/32, it is computationally expensive due to the additional grouping process, whereas CC balances both efficiency and effectiveness.

**Calibration on Long-Tail Datasets**  On long-tail datasets like ImageNet-LT, TS-based models struggle to provide effective calibration, especially for transformers. For example, on ViT-B/32, TS-based methods fail to calibrate effectively, as they apply uniform adjustments across the dataset, smoothing or sharpening probabilities globally. In contrast, CC and GC perform well on long-tail datasets, particularly with transformers. GC excels due to its multicalibration (Hébert-Johnson et al., 2018), offering sample-wise adjustments, though it comes at a high computational cost. By leveraging local uncertainty estimation through input perturbations, CC better captures uncertainties in underrepresented tail classes, making it especially useful for handling imbalanced data scenarios.

### 3.3 CALIBRATION PERFORMANCE ON OTHER METRICS

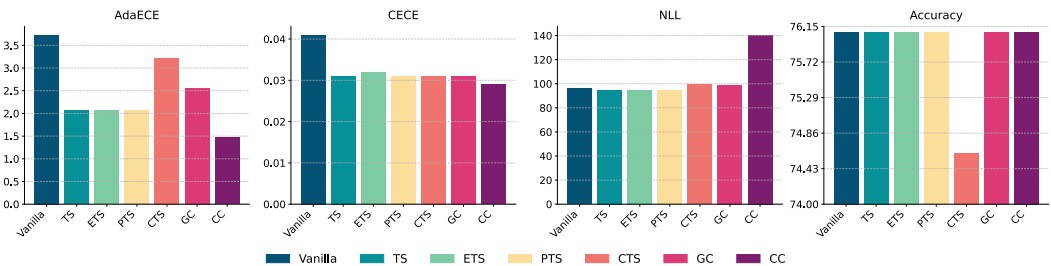

Figure 5: **Calibration performance of ResNet-50 on ImageNet-1K using AdaECE↓, CECE↓, NLL↓, and Accuracy↑.** ECE, AdaECE, and CECE are reported with 15 bins. Colors in the legend represent different methods. Results are averaged over 5 runs.

We also evaluate CC using additional metrics: AdaECE, CECE, NLL, and accuracy to provide a comprehensive view of its performance. Results for ResNet-50 on ImageNet are shown here, with results for other models and datasets available in Appendix C.

**AdaECE and CECE**  CC demonstrates superior performance on both AdaECE and CECE compared to traditional methods. AdaECE accounts for uneven confidence distributions, improving the reliability of ECE, while CECE gives detailed insights into classwise calibration. CC's strong results on both metrics show its effectiveness from different perspectives.

**Accuracy Maintained**  CC preserves the accuracy of the base models, showing no significant reduction in classification performance. As a post-hoc method, it does not require retraining, maintaining predictive capabilities, making it practical for real-world applications.

**Increase in NLL**  Interestingly, CC results in higher NLL values compared to other methods, reflecting a trade-off between calibration and the sharpness of probability estimates. This suggests that while CC reduces overconfidence in incorrect predictions, it also moderates overconfidence in correct predictions, leading to improved calibration without affecting accuracy.

| Dataset | Model | Cross-Entropy | | Brier Loss | | MMCE | | LS-0.05 | | FLSD-53 | | FL-3 | |
|---|---|---|---|---|---|---|---|---|---|---|---|---|---|
| | | base | ours | base | ours | base | ours | base | ours | base | ours | base | ours |
| CIFAR-10 | ResNet-50 | 4.34 | **0.78** | 1.80 | **1.07** | 4.56 | **0.83** | 2.97 | **1.24** | 1.55 | **0.49** | 1.48 | **0.66** |
| | ResNet-110 | 4.41 | **0.98** | 2.57 | **0.48** | 5.08 | **1.17** | **2.09** | 2.30 | 1.88 | **0.67** | 1.54 | **0.48** |
| | DenseNet-121 | 4.51 | **1.07** | 1.52 | **0.78** | 5.10 | **1.18** | 1.87 | **1.39** | 1.23 | **0.68** | 1.31 | **0.98** |
| | Wide-ResNet | 3.24 | **0.36** | 1.24 | **0.58** | 3.29 | **0.39** | 4.25 | **1.15** | 1.58 | **0.49** | 1.68 | **0.53** |
| CIFAR-100 | ResNet-50 | 17.52 | **1.25** | 6.57 | **1.57** | 15.32 | **1.98** | 7.82 | **5.08** | 4.49 | **1.43** | 5.16 | **1.52** |
| | ResNet-110 | 19.05 | **4.57** | 7.88 | **3.24** | 19.14 | **4.41** | 11.04 | **4.58** | 8.55 | **3.47** | 8.64 | **3.67** |
| | DenseNet-121 | 20.99 | **5.40** | 5.22 | **1.82** | 19.10 | **3.76** | 12.87 | **4.99** | 3.70 | **1.41** | 4.14 | **1.94** |
| | Wide-ResNet | 15.34 | **1.61** | 4.34 | **1.87** | 13.17 | **2.17** | 4.89 | **4.21** | 3.02 | **1.64** | 2.14 | **1.78** |

Table 2: **Comparison of Train-time Calibration Methods Using ECE↓ Across Various Datasets and Models.** ECE values are reported with 15 bins. The best-performing method for each dataset-model combination is in bold, and our method (CC) is highlighted. Results are averaged over 5 runs.

### 3.4 COMPARISON WITH TRAINING-TIME CALIBRATION METHODS

We evaluate CC alongside training-time calibration techniques, including Brier Loss (Brier, 1950), Maximum Mean Calibration Error (MMCE) (Kumar et al., 2018), Label Smoothing (LS-0.05) (Szegedy et al., 2016), and Focal Loss variants (FLSD-53 and FL-3) (Mukhoti et al., 2020), as

shown in Table 2. Our analysis shows that combining CC with these methods consistently enhances calibration performance across various models and datasets, further validating CC's effectiveness alongside training-time approaches.

Moreover, as seen in Table 1, CC alone, as a post-hoc calibration method, already outperforms these train-time techniques with minimal computational overhead, while train-time methods require significantly more resources. Additional results for other settings are available in Appendix D.

### 3.5 ABLATION STUDY

**Aggregation Methods**  In our ablation study, we compare two aggregation methods for refining confidence estimates: the mean of softmax probabilities (*Mean*), defined as:

$$\hat{p}_k = \frac{1}{T} \sum_{t=1}^{T} \text{softmax}\left( q\left( \widetilde{h(x)}^t \right) \right), \quad \text{for } k = 1, \dots, K, \tag{10}$$

and consistency-based aggregation (*Consis.*) as shown in Eq. 9. Both methods leverage predictions over perturbed logits. The mean of softmax probabilities treats the perturbation process like an ensemble method, interpreting uncertainty as a distribution. We show the evaluation results on CIFAR-10 and CIFAR-100 in Table 3. On smaller datasets like CIFAR-10, both methods perform similarly. However, on larger datasets with more classes, such as CIFAR-100 and ImageNet, consistency-based aggregation slightly outperforms softmax averaging. This suggests that consistency-based aggregation captures uncertainty better than the view of ensemble.

**Choice of Noise**  We investigate the impact of different noise types for input perturbations, comparing uniform noise (*U*) and Gaussian noise (*G*), as shown in Table 3. Uniform noise performs better on datasets with fewer classes, such as CIFAR-10 and CIFAR-100. However, on larger datasets like ImageNet, Gaussian noise yields better results, likely due to variations in the gap between the maximum and second maximum logits across datasets as shown in Figure 4. The choice of noise is treated as a hyperparameter, offering flexibility to adapt to different datasets and models.

**Number of Perturbations**  We also assess the impact of the number of perturbations. As shown in Figure 4, our experiments indicate that CC achieves strong calibration performance with as few as $2^4 = 16$ perturbations. Although increasing the number of perturbations slightly improves results, the diminishing returns suggest that CC provides robust calibration with a moderate number of perturbations, ensuring both efficiency and accuracy.

| Dataset | Model | Mean U | Mean G | Consis. U | Consis. G |
|---------|-------|--------|--------|-----------|-----------|
| CIFAR-10 | ResNet-50 | **0.72** | 1.34 | 0.78 | 1.33 |
| | Wide-ResNet | 0.37 | 0.80 | **0.36** | 0.83 |
| CIFAR-100 | ResNet-50 | 1.52 | 2.70 | **1.25** | 2.49 |
| | Wide-ResNet | 1.86 | 2.08 | **1.61** | 1.88 |
| ImageNet | ResNet-50 | 2.37 | 1.41 | 2.29 | **1.27** |
| | Wide-ResNet-50 | 2.17 | 1.7 | 2.23 | **1.57** |

Table 3: **Comparison of Aggregation Methods and Noise Types Using ECE↓ Across Various Datasets and Models.** ECE values are reported using 15 bins. The best-performing method for each dataset-model combination is highlighted in bold.

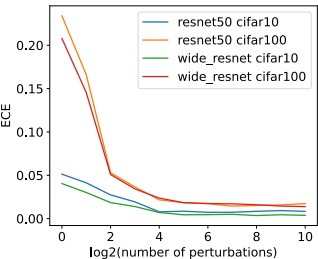

Table 4: **Effect of Number of Perturbations on Calibration Performance (ECE↓)**

## 4 CONCLUSION

Consistency offers an alternative perspective on calibration by focusing on prediction stability under perturbations as an indicator of confidence. CC has proven highly effective in reducing calibration errors across various datasets. However, CC has limitations, such as the need for tuning perturbation strength and noise type, and its current focus on classification tasks, with its application to regression remaining unexplored. Future work can aim to develop a new, more universal consistency-based metric to complement existing metrics like ECE. This would provide a more comprehensive evaluation to calibration, ultimately leading to more reliable deep learning models.

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
