## A    RELATED WORKS

**Calibration Methods**    Post-hoc calibration methods adjust model outputs after training to improve calibration. A widely used technique is Temperature Scaling (TS) (Guo et al., 2017), which smooths softmax probabilities by search a temperature factor on a validation set. Enhanced variants of TS include Parameterized Temperature Scaling (PTS) (Tomani et al., 2022), which uses a neural network to learn the temperature, and Class-based Temperature Scaling (CTS) (Frenkel et al., 2021), which applies adjustments on a class-wise basis. Group Calibration (GC) (Yang et al., 2024) and ProCal (Xiong et al., 2023a) aim for multi-calibration (Hébert-Johnson et al., 2018) by splitting data samples by proximity and grouping. Another stream of work is train-time calibration such as Brier Loss (Brier, 1950), Dirichlet Scaling (Kull et al., 2019), Maximum Mean Calibration Error (MMCE) (Kumar et al., 2018), Label Smoothing (Szegedy et al., 2016), and Focal Loss (Mukhoti et al., 2020) and Dual Focal Loss (Tao et al., 2023). However, these methods often require substantial higher computational overhead.

**Ensemble-Based Calibration**    Ensemble-based methods ensemble multiple outputs in different ways. They use models or samples to approximate Bayesian Inference. Lakshminarayanan et al. (2017) propose deep ensembles as a scalable alternative to Bayesian Neural Networks (BNNs) for uncertainty estimation. Similarly, Gal & Ghahramani (2016) treat dropout as approximate Bayesian inference. Data-centric ensemble techniques using test-time augmentation, as described by Conde et al. (2023), also help improve calibration. Zhang et al. (2020) resort to the power of Bayesian inference and proposed a Ensemble-based TS (ETS). However, these methods typically require significant computational resources to train multiple models or perform repeated inferences. In contrast, our approach relies on consistency rather than probability distribution modeling.

**Consistency in LLMs**    Consistency has emerged as a key approach for black-box uncertainty estimation and hallucination detection in large language models (LLMs). These methods evaluate uncertainty by measuring variability in outputs across slight changes, such as different sampling techniques or rephrased prompts. Confident models produce stable outputs, while variability indicates uncertainty. For instance, SelfCheckGPT (Manakul et al., 2023) uses sampling and similarity metrics like BERTScore and NLI to detect hallucinations, while Lin et al. (2023) analyze a similarity matrix to estimate uncertainty. Xiong et al. (2023b) further break down uncertainty estimation into prompting, sampling, and consistency-based aggregation. These methods, which rely on output stability, are efficient alternatives to probabilistic approaches.

## B    PERTURBATION OF DIFFERENT LAYER

This section presents a detailed analysis of the impact of perturbations applied at various levels of a ResNet50 model, trained on CIFAR-10. The experiments were conducted using 32 samples, and the effects on ECE, accuracy, and optimal perturbation values were evaluated.

| Perturbation Level | ECE (%) | Accuracy (%) | Optimal Perturbation |
|---|---|---|---|
| Image | 1.1 | 95.25 | train aug jitter0.1 |
| Logits | 0.73 | 95.04 | 8.2 |
| Feature (Last Layer) | 2.06 | 95.06 | 3.0 |
| Feature (Layer 4) | 0.53 | 95.29 | 13.28 |
| Feature (Layer 3) | 53.12 | 10.03 | 20.12 |
| Feature (Layer 2) | 56.28 | 10.02 | 20.21 |
| Feature (Layer 1) | 49.53 | 10.11 | 20.75 |

Table 5: Comparison of perturbations at different layers with number of samples set to 32 using ECE↓ and Accuracy↑, evaluated on ResNet50 with CIFAR-10. ECE values are reported with 15 bins. Optimal Perturbations for logits and features are represented in $\epsilon$ value

From Table 5, we observe a clear trend in the performance of perturbations applied at different layers of the model. Perturbation at the logits level achieves a favorable trade-off between calibration and efficiency. Although the perturbation applied to the fourth layer's feature space slightly improves the ECE to 0.53%, the associated computational cost is significantly higher, with the optimal perturbation value of 13.28.

On the other hand, perturbations applied at lower feature levels (Layer 1 to Layer 3) result in severe degradation of both accuracy and calibration. Specifically, the ECE increases drastically to above 50%, and accuracy drops to approximately 10%, with a significant increase in computing time and memory use. This suggests that perturbing the features at these lower layers disrupts the model's ability to recognize patterns and correctly classify the input data. We hypothesize that this is due to the higher sensitivity of lower layers to the raw data structure, where perturbations may significantly distort the features necessary for effective recognition.

## C COMPARISON OF POST-HOC CALIBRATION METHODS ON OTHER METRICS

As shown in table 6, The proposed CC method consistently achieves the lowest AdaECE values, outperforming the other methods. This indicates better calibration performance, in line with our discussion in the main text. For instance, in CIFAR-10, Wide-ResNet has an AdaECE of 0.40 with CC compared to 3.24 for Vanilla, showing a significant improvement. Similar results are observed across other models and datasets. The formula for Adaptive-ECE is as follows:

$$\text{Adaptive-ECE} = \sum_{i=1}^{B} \frac{|B_i|}{N} |I_i - C_i| \ \text{ s.t. } \forall i, j \cdot |B_i| = |B_j| \tag{11}$$

| Dataset | Model | Vanilla | TS | ETS | PTS | CTS | GC | CC (ours) |
|---------|-------|---------|-----|-----|-----|-----|-----|-----------|
| CIFAR-10 | ResNet-50 | 4.33 | 2.14 | 2.14 | 2.14 | 1.71 | 1.24 | **0.64** |
| | ResNet-110 | 4.40 | 1.89 | 1.89 | 1.90 | 1.31 | **0.94** | 0.96 |
| | DenseNet-121 | 4.49 | 2.12 | 2.12 | 2.12 | 1.71 | 1.28 | **1.20** |
| | Wide-ResNet | 3.24 | 1.71 | 1.71 | 1.71 | 1.42 | 1.17 | **0.40** |
| CIFAR-100 | ResNet-50 | 17.52 | 5.76 | 5.72 | 5.66 | 5.79 | 3.43 | **1.61** |
| | Wide-ResNet | 15.34 | 4.48 | 4.45 | 4.41 | 4.69 | 2.24 | **1.73** |
| ImageNet | ResNet-50 | 3.73 | 2.07 | 2.07 | 2.06 | 3.22 | 2.56 | **1.47** |
| | DenseNet-121 | 6.59 | 1.67 | 1.68 | 1.69 | 1.89 | 2.49 | **1.36** |
| | Wide-ResNet-50 | 5.32 | 2.97 | 2.97 | 2.95 | 4.13 | 2.18 | **1.27** |
| | ViT-B-16 | 5.59 | 4.05 | 4.06 | 4.08 | 5.50 | 1.86 | **1.76** |
| | ViT-B-32 | 6.40 | 3.83 | 3.85 | 3.91 | 5.73 | **1.33** | 1.77 |

Table 6: **Comparison of Post-Hoc Calibration Methods Using AdaECE↓ Across Various Datasets and Models.** AdaECE values are reported with 15 bins. The best results for each combination is in bold, and our method (CC) is highlighted. Results are averaged over 5 runs.

As shown in table 7, The CC method also performs the best in terms of class-wise calibration, with consistently lower CECE values. This confirms that CC provides better calibration across individual classes, as discussed in the main body. For example, for ResNet-50 on CIFAR-100, CC achieves a CECE of 0.20, which is the lowest among the methods. CECE is another measure of calibration performance that addresses the deficiency of ECE in only measuring the calibration performance of the single predicted class. It can be formulated as:

$$\text{Classwise-ECE} = \frac{1}{\mathcal{K}} \sum_{i=1}^{B} \sum_{j=1}^{\mathcal{K}} \frac{|B_{i,j}|}{N} |I_{i,j} - C_{i,j}| \tag{12}$$

As shown in table 8, interestingly, the NLL values are generally higher with the CC method compared to some other calibration methods, despite its superior calibration performance in AdaECE and CECE. This suggests that while CC improves calibration, it may come at the cost of slightly higher NLL values. For instance, for CIFAR-100 on ResNet-50, CC has a higher NLL than TS, but it remains competitive overall.

9 indicates that there is little to no change in accuracy across the calibration methods, with all methods performing similarly in terms of classification accuracy. This patter is consistent with the main section, showing CC improves calibration without sacrificing accuracy. For example, on CIFAR-10, Wide-ResNet achieves almost identical accuracy for all methods, with CC slightly outperforming others in specific cases.

| Dataset | Model | Vanilla | TS | ETS | PTS | CTS | GC | CC (ours) |
|---------|-------|---------|-----|-----|-----|-----|-----|-----------|
| CIFAR-10 | ResNet-50 | 0.91 | 0.45 | 0.45 | 0.45 | 0.41 | 0.46 | **0.39** |
| | ResNet-110 | 0.92 | 0.48 | 0.48 | 0.48 | 0.42 | 0.52 | **0.41** |
| | DenseNet-121 | 0.92 | 0.48 | 0.48 | 0.48 | **0.41** | 0.54 | 0.43 |
| | Wide-ResNet | 0.68 | 0.37 | 0.37 | 0.37 | 0.37 | 0.48 | **0.32** |
| CIFAR-100 | ResNet-50 | 0.38 | 0.21 | 0.21 | 0.21 | 0.22 | 0.21 | **0.20** |
| | Wide-ResNet | 0.34 | 0.19 | 0.19 | 0.19 | 0.20 | 0.20 | **0.18** |
| ImageNet | ResNet-50 | **0.03** | **0.03** | **0.03** | **0.03** | **0.03** | **0.03** | 0.03 |
| | DenseNet-121 | **0.03** | **0.03** | **0.03** | **0.03** | **0.03** | **0.03** | 0.03 |
| | Wide-ResNet-50 | 0.03 | 0.03 | 0.03 | 0.03 | 0.03 | 0.03 | **0.02** |
| | ViT-B-16 | 0.03 | **0.02** | **0.02** | **0.02** | 0.03 | **0.02** | **0.02** |
| | ViT-B-32 | **0.03** | **0.03** | **0.03** | **0.03** | **0.03** | **0.03** | 0.03 |

Table 7: **Comparison of Post-Hoc Calibration Methods Using CECE↓ Across Various Datasets and Models.** CECE values are reported with 15 bins. The best-performing method for each dataset-model combination is in bold, and our method (CC) is highlighted. Results are averaged over 5 runs.

| Dataset | Model | Vanilla | TS | ETS | PTS | CTS | GC | CC (ours) |
|---------|-------|---------|-----|-----|-----|-----|-----|-----------|
| CIFAR-10 | ResNet-50 | 41.21 | 20.39 | 20.39 | 20.38 | 20.15 | **19.97** | 20.39 |
| | ResNet-110 | 47.52 | 21.52 | 21.52 | 21.52 | 20.84 | **20.68** | 23.33 |
| | DenseNet-121 | 42.93 | 21.78 | 21.78 | 21.78 | 21.01 | **20.30** | 22.19 |
| | Wide-ResNet | 26.75 | 15.33 | 15.33 | 15.33 | **15.13** | 15.32 | 17.10 |
| CIFAR-100 | ResNet-50 | 153.67 | **106.07** | **106.07** | **106.07** | 106.25 | 107.80 | 108.40 |
| | Wide-ResNet | 140.11 | **95.71** | **95.71** | **95.71** | 96.38 | 96.92 | 99.30 |
| ImageNet | ResNet-50 | 96.12 | 94.82 | 94.82 | **94.81** | 99.58 | 99.07 | 140.57 |
| | DenseNet-121 | 109.52 | **103.90** | **103.90** | 103.91 | 106.13 | 108.14 | 162.02 |
| | Wide-ResNet-50 | 88.56 | **86.46** | **86.46** | **86.46** | 91.68 | nan | 120.59 |
| | ViT-B-16 | 83.71 | 78.63 | 78.63 | **78.63** | 85.19 | 82.14 | 106.89 |
| | ViT-B-32 | 107.76 | 101.67 | 101.67 | **101.66** | 107.53 | 105.45 | 141.71 |

Table 8: **Comparison of Post-Hoc Calibration Methods Using NLL↓ Across Various Datasets and Models.** The best-performing method for each dataset-model combination is in bold, and our method (CC) is highlighted. Results are averaged over 5 runs.

In figure 6, we see that the proposed CC method significantly reduces both AdaECE and CECE values compared to other calibration methods, indicating better calibration for Wide-ResNet on CIFAR-10. The accuracy remains mostly unchanged across all methods, while NLL is slightly higher for CC compared to other methods like TS and ETS. This behavior is consistent with our findings in the main text.

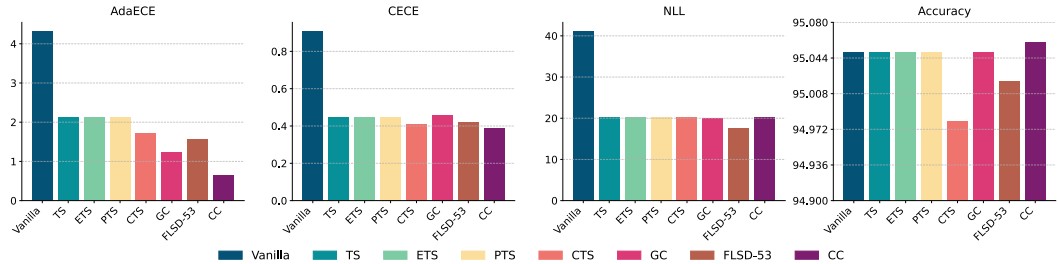

Figure 6: **Calibration performance of ResNet-50 on Cifar-10 using AdaECE↓, CECE↓, NLL↓, and Accuracy↑.** ECE, AdaECE, and CECE are reported with 15 bins. Colors in the legend represent different methods. Results are averaged over 5 runs.

In Figure 6, for ResNet-50 on CIFAR-10, the CC method demonstrates excellent performance with the lowest AdaECE and CECE values, further supporting its effectiveness in calibration. NLL is higher for CC, which is interesting given its superior performance in other metrics. However, accuracy remains largely unchanged, consistent with the overall findings discussed in the text.

Figure 8 illustrates the performance of ResNet-50 on CIFAR-100 across different calibration methods. The proposed CC method again shows the lowest AdaECE and CECE, confirming its superior

| Dataset | Model | Vanilla | TS | ETS | PTS | CTS | GC | CC (ours) |
|---------|-------|---------|-----|-----|-----|-----|-----|-----------|
| CIFAR-10 | ResNet-50 | 95.05 | 95.05 | 95.05 | 95.05 | 94.98 | 95.05 | 95.06 |
| | ResNet-110 | 95.11 | 95.11 | 95.11 | 95.11 | **95.18** | 95.11 | 95.16 |
| | DenseNet-121 | 95.02 | 95.02 | 95.02 | 95.02 | 95.01 | 95.02 | 95.04 |
| | Wide-ResNet | **96.13** | **96.13** | **96.13** | **96.13** | 96.06 | **96.13** | **96.13** |
| CIFAR-100 | ResNet-50 | 76.70 | 76.70 | 76.70 | 76.70 | **76.72** | 76.70 | 76.71 |
| | Wide-ResNet | 79.29 | 79.29 | 79.29 | 79.29 | 79.17 | 79.29 | **79.31** |
| ImageNet | ResNet-50 | **76.08** | **76.08** | **76.08** | **76.08** | 74.62 | **76.08** | **76.08** |
| | DenseNet-121 | 74.16 | 74.16 | 74.16 | 74.16 | 73.08 | 74.16 | **74.37** |
| | Wide-ResNet-50 | 78.40 | 78.40 | 78.40 | 78.40 | 77.07 | 78.40 | **78.48** |
| | ViT-B-16 | **81.09** | **81.09** | **81.09** | **81.09** | 80.01 | **81.09** | 81.06 |
| | ViT-B-32 | **75.94** | **75.94** | **75.94** | **75.94** | 74.90 | **75.94** | 75.90 |

Table 9: **Comparison of Post-Hoc Calibration Methods Using Accuracy↑ Across Various Datasets and Models.** Top-1 accuracy values are reported. The best results for each combination is in bold, and our method (CC) is highlighted. Results are averaged over 5 runs.

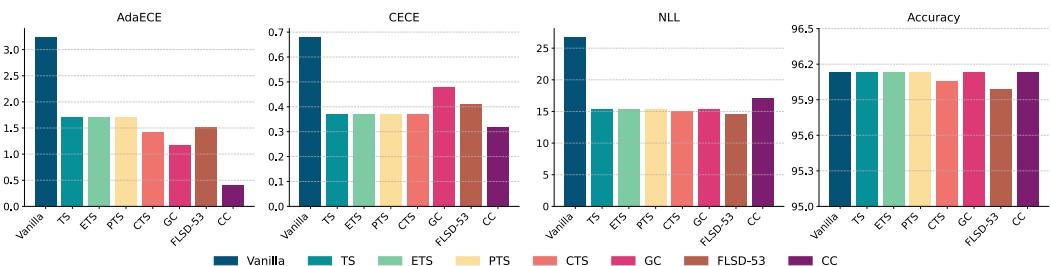

Figure 7: **Calibration performance of Wide-ResNet on CIFAR-10 using AdaECE↓, CECE↓, NLL↓, and Accuracy↑.** ECE, AdaECE, and CECE are reported with 15 bins. Colors in the legend represent different methods. Results are averaged over 5 runs.

calibration performance. NLL for CC is slightly higher compared to TS, but accuracy shows minimal changes across methods. These results align with our overall conclusions that CC improves calibration without sacrificing accuracy.

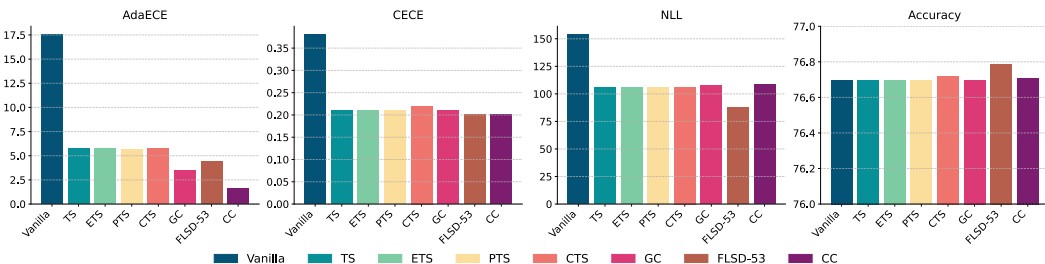

Figure 8: **Calibration performance of ResNet-50 on CIFAR-100 using AdaECE↓, CECE↓, NLL↓, and Accuracy↑.** ECE, AdaECE, and CECE are reported with 15 bins. Colors in the legend represent different methods. Results are averaged over 5 runs.

# D COMPARISON OF VARIOUS TRAINING-TIME CALIBRATION METHODS ON OTHER METRICS

As shown in Table 10, CC consistently outperforms baseline models across all metrics and datasets. Specifically, on CIFAR-10 and CIFAR-100, CC achieves significantly lower AdaECE scores for ResNet-50, ResNet-110, DenseNet-121, and Wide-ResNet compared to traditional methods such as Brier Loss, and MMCE. For instance, on CIFAR-100 with ResNet-110, CC reduces the AdaECE from 19.05 (baseline) to 5.28, showing superior calibration performance.

| Dataset | Model | Cross-Entropy | | Brier Loss | | MMCE | | LS-0.05 | | FLSD-53 | | FL-3 | |
|---|---|---|---|---|---|---|---|---|---|---|---|---|---|
| | | base | ours | base | ours | base | ours | base | ours | base | ours | base | ours |
| CIFAR-10 | ResNet-50 | 4.33 | **0.64** | 1.75 | **0.99** | 4.55 | **1.06** | 3.88 | **1.74** | 1.56 | **0.36** | 1.95 | **0.71** |
| | ResNet-110 | 4.40 | **0.96** | 2.60 | **0.30** | 5.07 | **1.80** | 4.48 | **2.43** | 2.08 | **0.73** | 1.64 | **0.38** |
| | DenseNet-121 | 4.49 | **1.20** | 2.02 | **0.64** | 5.10 | **1.76** | 4.40 | **1.94** | 1.38 | **0.53** | 1.23 | **0.69** |
| | Wide-ResNet | 3.24 | **0.40** | 1.70 | **0.57** | 3.29 | **0.63** | 4.27 | **1.54** | 1.52 | **0.42** | 1.84 | **0.42** |
| CIFAR-100 | ResNet-50 | 17.52 | **1.61** | 6.55 | **1.90** | 15.32 | **1.88** | 7.66 | **6.17** | 4.39 | **1.48** | 5.09 | **1.70** |
| | ResNet-110 | 19.05 | **5.28** | 7.72 | **3.54** | 19.14 | **5.14** | 11.14 | **8.00** | 8.56 | **3.50** | 8.64 | **3.98** |
| | DenseNet-121 | 20.99 | **5.85** | 5.04 | **2.02** | 19.10 | **3.90** | 12.83 | **7.06** | 3.54 | **1.52** | 4.14 | **2.03** |
| | Wide-ResNet | 15.34 | **1.73** | 4.28 | **1.92** | 13.16 | **2.06** | 5.14 | **4.75** | 2.77 | **1.79** | 2.07 | **1.58** |

Table 10: **Comparison of Train-time Calibration Methods Using AdaECE↓ Across Various Datasets and Models.** AdaECE values are reported with 15 bins. The best results for each combination is in bold, and our method (CC) is highlighted. Results are averaged over 5 runs.

In Table 11, the CECE results further reinforce the effectiveness of CC across all metrics. For CIFAR-10, CC improves CECE for all models compared to baseline methods. For instance, with ResNet-50, the CECE decreases from 0.91 to 0.39. Similar trends are observed on CIFAR-100, with Wide-ResNet showing a reduction in CECE from 0.34 (baseline) to 0.18 when using CC, demonstrating enhanced class-wise calibration.

| Dataset | Model | Cross-Entropy | | Brier Loss | | MMCE | | LS-0.05 | | FLSD-53 | | FL-3 | |
|---|---|---|---|---|---|---|---|---|---|---|---|---|---|
| | | base | ours | base | ours | base | ours | base | ours | base | ours | base | ours |
| CIFAR-10 | ResNet-50 | 0.91 | **0.39** | 0.46 | **0.35** | 0.94 | **0.47** | 0.71 | **0.53** | 0.42 | **0.35** | 0.43 | **0.39** |
| | ResNet-110 | 0.92 | **0.41** | 0.59 | **0.41** | 1.04 | **0.50** | **0.66** | 0.67 | 0.48 | **0.39** | 0.43 | **0.37** |
| | DenseNet-121 | 0.92 | **0.43** | 0.46 | **0.37** | 1.04 | **0.59** | 0.60 | **0.48** | 0.41 | **0.35** | 0.42 | **0.35** |
| | Wide-ResNet | 0.68 | **0.32** | 0.44 | **0.32** | 0.70 | **0.38** | 0.79 | **0.41** | 0.41 | **0.28** | 0.44 | **0.30** |
| CIFAR-100 | ResNet-50 | 0.38 | **0.20** | 0.22 | **0.19** | 0.34 | **0.18** | 0.23 | **0.22** | 0.20 | **0.19** | 0.20 | **0.19** |
| | ResNet-110 | 0.41 | **0.21** | 0.24 | **0.19** | 0.42 | **0.20** | 0.26 | **0.22** | 0.24 | **0.19** | 0.24 | **0.20** |
| | DenseNet-121 | 0.45 | **0.23** | 0.20 | **0.20** | 0.42 | **0.23** | 0.29 | **0.22** | 0.19 | **0.19** | 0.20 | **0.19** |
| | Wide-ResNet | 0.34 | **0.18** | 0.19 | **0.18** | 0.30 | **0.17** | 0.21 | **0.19** | 0.18 | **0.17** | 0.18 | **0.17** |

Table 11: **Comparison of Train-time Calibration Methods Using CECE↓ Across Various Datasets and Models.** CECE values are reported with 15 bins. The best results for each combination is in bold, and our method (CC) is highlighted. Results are averaged over 5 runs.

Table 12 presents the NLL comparison. It is interesting as mentioned in the main section, the CC method sometimes produces higher NLL values.

| Dataset | Model | Cross-Entropy | | Brier Loss | | MMCE | | LS-0.05 | | FLSD-53 | | FL-3 | |
|---|---|---|---|---|---|---|---|---|---|---|---|---|---|
| | | Base | Ours | Base | Ours | Base | Ours | Base | Ours | Base | Ours | Base | Ours |
| CIFAR-10 | ResNet-50 | 41.2 | **20.4** | **18.7** | 22.3 | 44.8 | **20.9** | **27.7** | 29.3 | **17.6** | 22.7 | **18.4** | 24.2 |
| | ResNet-110 | 47.5 | **25.5** | **20.4** | 22.5 | 55.7 | **25.5** | 29.9 | **29.4** | **18.5** | 21.9 | **17.8** | 23.1 |
| | DenseNet-121 | 42.9 | **24.0** | **19.1** | 21.2 | 52.1 | **31.2** | 28.7 | **28.5** | **18.4** | 27.2 | **18.0** | 28.3 |
| | Wide-ResNet | 26.8 | **17.1** | **15.9** | 16.2 | 28.5 | **18.2** | 21.7 | 24.5 | **14.6** | 17.6 | **15.2** | 19.9 |
| CIFAR-100 | ResNet-50 | 153.7 | **113.0** | **99.6** | 133.5 | 125.3 | **116.7** | 121.0 | 133.9 | **88.0** | 128.8 | **87.5** | 128.1 |
| | ResNet-110 | 179.2 | **122.3** | **110.7** | 146.9 | 180.6 | **125.3** | 133.1 | 141.4 | **89.9** | 126.9 | **90.9** | 132.0 |
| | DenseNet-121 | 205.6 | **163.1** | **98.3** | 139.9 | 166.6 | **146.8** | 142.0 | 185.8 | **85.5** | 129.0 | **87.1** | 130.8 |
| | Wide-ResNet | 140.1 | **102.5** | **84.6** | 98.7 | 119.6 | **109.3** | 108.1 | 136.6 | **76.9** | 108.7 | **74.7** | 106.8 |

Table 12: **Comparison of Train-time Calibration Methods Using NLL↓ Across Various Datasets and Models.** The best-performing method for each dataset-model combination is in bold, and our method (CC) is highlighted. Results are averaged over 5 runs.

Table 13 presents a comparison of classification accuracies. While achieving superior calibration performance by CC, the accuracy remains unaffected across all metrics.

| Dataset | Model | Cross-Entropy | | Brier Loss | | MMCE | | LS-0.05 | | FLSD-53 | | FL-3 | |
|---|---|---|---|---|---|---|---|---|---|---|---|---|---|
| | | base | ours | base | ours | base | ours | base | ours | base | ours | base | ours |
| CIFAR-10 | ResNet-50 | **95.05** | 95.06 | **94.99** | 95.01 | 95.01 | **94.99** | 94.71 | **94.68** | 95.02 | **94.95** | 94.75 | 94.75 |
| | ResNet-110 | **95.11** | 95.16 | 94.52 | **94.48** | **94.60** | 94.63 | **94.48** | 94.49 | **94.57** | 94.63 | **94.92** | 94.94 |
| | DenseNet-121 | 95.02 | **95.01** | 94.90 | **94.86** | **94.59** | 94.60 | 94.91 | 94.91 | 94.58 | **94.51** | 94.66 | 94.66 |
| | Wide-ResNet | 96.13 | **96.12** | 95.92 | **95.90** | 96.09 | **96.05** | 95.80 | 95.83 | 95.99 | 96.01 | 95.87 | 95.87 |
| CIFAR-100 | ResNet-50 | **76.70** | 76.71 | 76.60 | **76.58** | 76.80 | 76.80 | **76.56** | 76.65 | 76.79 | **76.73** | 77.24 | 77.34 |
| | ResNet-110 | 77.27 | **77.17** | 74.91 | **74.79** | **76.93** | 76.96 | **76.57** | 76.64 | **77.48** | 77.49 | 77.08 | **77.04** |
| | DenseNet-121 | **75.47** | 75.49 | **76.27** | 76.30 | 76.03 | 76.03 | **75.94** | 75.96 | 77.34 | 77.34 | **76.76** | 76.85 |
| | Wide-ResNet | 79.29 | **79.25** | 79.43 | **79.29** | 79.27 | **79.23** | 78.83 | 78.88 | **79.91** | 79.92 | **80.30** | 80.34 |

Table 13: **Comparison of Train-time Calibration Methods Using Accuracy↑ Across Various Datasets and Models.** Top-1 Accuracy values are reported. The best results for each combination is in bold, and our method (CC) is highlighted. Results are averaged over 5 runs.