# OpenReview forum: "Consistency Calibration: Improving Uncertainty Calibration via Consistency among Perturbed Neighbors"
_ICLR.cc/2025/Conference — ICLR 2025 Conference Withdrawn Submission_

### Official Review · Reviewer_u8As · 2024-11-03

**Soundness:** 3
**Presentation:** 3
**Contribution:** 2
**Rating:** 3
**Confidence:** 2

**Summary:**

This paper proposes a new approach called Consistency Calibration (CC) to improve uncertainty calibration in deep learning models. Instead of the traditional reliability-based calibration methods like Expected Calibration Error (ECE), the authors introduce a novel perspective called consistency, which evaluates the model's confidence by measuring whether predictions remain stable across perturbed inputs. The effectiveness of CC is validated on various datasets, including CIFAR-10, CIFAR-100, ImageNet, and ImageNet-LT.

**Strengths:**

Introduction of a New Calibration Perspective: This paper introduces consistency as a new criterion for calibration, providing a potentially more reliable alternative to traditional methods of evaluating model confidence.
Efficient Implementation: Consistency Calibration uses input perturbations to measure uncertainty, requiring no additional data or label information, which allows for a simple and computationally efficient post-hoc calibration process.

**Weaknesses:**

Lack of Scalability Validation for Large Models: While the proposed method demonstrates superior performance on several datasets, there is a lack of validation on larger models, such as massive language models or more complex real-world scenarios. Future work should include experiments to assess the scalability of CC in such settings.
Need for Optimization of Perturbation Strength and Noise Type: Consistency Calibration requires optimization of the perturbation strength and noise type, which adds hyperparameter tuning complexity. To address this issue, automated hyperparameter tuning techniques should be explored to enhance practical usability.
Limited Experimental Scope for Calibration Performance: The experiments are focused primarily on key image datasets, with no validation in other domains (e.g., natural language processing, speech recognition). Further research should evaluate the performance of Consistency Calibration across various domains to establish its broader applicability.

**Questions:**

see weakness

---

### Official Review · Reviewer_bUZw · 2024-11-03

**Soundness:** 3
**Presentation:** 2
**Contribution:** 2
**Rating:** 5
**Confidence:** 3

**Summary:**

This paper proposes a method called consistency calibration, which uses perturbed logits for calibration. Consistency under perturbation is treated as an indicator of confidence, aiming to improve the model's calibration.

**Strengths:**

- The proposed consistency calibration method is interesting.
- The writing is generally clear.
- The experiments are conducted across CNN and ViTs over cifar10, 100, ImageNet and ImageNet-LT.

**Weaknesses:**

- Some results or interpretations are unclear
   - Figure 1(a) is missing "Moderate Augmentation" in the legend. Does the accuracy of Train Augmentation go below 92%? Adding detailed numerical results would help clarify.
   - The final results use CC as a post-hoc method, but Sec 2.5 recommends training-time augmentation, which seems inconsistent.
   - Figure 3(c) only shows a case study for comparing CC, CC (Train Aug), and CC (Train Aug + Jitter). It would help to include more performance comparisons between these methods.

- The analysis of why consistency calibration work is not sufficient. The larger logit gap between correct and incorrect samples might help maintain accuracy under logit perturbation, but it doesn’t fully explain the better calibration. A more rigorous or theoretical analysis is needed.

**Questions:**

- Are ECE and AdaECE calculated on the original test set or the perturbed set? Lines 229-231 suggest it’s the perturbed set.
- The sensitivity comparison in Lines 266-269 is hard to assess since the scales differ. The error behavior outside the range 0.5-1 is also unclear. Plus, it’s uncertain if this analysis on 2D data would apply to high-dimensional data like images.

---

### Official Review · Reviewer_tFK1 · 2024-11-06

**Soundness:** 2
**Presentation:** 3
**Contribution:** 2
**Rating:** 3
**Confidence:** 5

**Summary:**

In the paper, the authors propose a method to calibrate image classifier probabilities using perturbations in the input image or hidden features, instead of other popular calibration methods such as various versions of Temperature Scaling. On several image classification benchmarks (e.g., CIFAR and ImageNet), the authors demonstrate that the final recalibrated predicted probabilities—generated by sampling labels from predicted class distributions for multiple perturbed image versions and then averaging them to recompute new class probabilities—can achieve low calibration errors (e.g., ECE) and outperform some calibration baselines.

**Strengths:**

* Topic model calibration is important within the domains of robustness, safety, and uncertainty, especially considering recent developments in deep learning.

* The paper is clearly written and easy to follow, with an intuitive and easily understandable idea. The related work section provides sufficient discussion of existing calibration metrics.

* The approach does not require architectural changes to the model.

**Weaknesses:**

* **the paper overlooks a significant body of relevant literature.** For example, the idea developed in [1] to generate uncertainty by running several inferences with perturbed inputs is very similar to the approach proposed in this paper, as are other sensitivity-based methods such as [2], which shares the same motivation as the proposed method, significantly undermining the paper's novelty. Additionally, [3], mentioned in the appendix, bears a strong resemblance to the proposed method. Given this, positioning the method as a novel approach to calibration or uncertainty estimation seems overstated.


* **the experiments lack relevant baselines.** Although the authors propose a version of the method where noise is added to intermediate hidden representations, the original formulation—where noise is applied to the inputs—requires multiple inferences. In this context, ensembling-based approaches (some of which are mentioned in the appendix) should be considered as relevant baselines and should not be omitted from the experiments. Moreover, it is unclear how the inference-time vs. calibration-quality trade-off is addressed in the proposed method; some existing ensembling approaches, such as Deep Ensembles, MC-Dropout, and BatchEnsembles, may require additional computation but consistently demonstrate the highest uncertainty quality [4] (in terms of both calibration and epistemic uncertainty). Additionally, recently introduced efficient ensembling approaches, such as PackedEnsembles [5], Depth Uncertainty [6], or MixMo [7], require only a single inference to generate multiple predictions and estimate ensembling-based uncertainty, thereby mitigating computational overhead while maintaining the high uncertainty quality of ensembles.


* related to the previous point, **the method is specifically designed with image classification in mind**, making both its design and the conducted experiments quite limited. The original formulation, which involves adding perturbations to the input, is not feasible for other popular domains, such as language modeling, where adding Gaussian noise is not feasible. Additionally, certain architectures, such as GNNs, are also not suitable for this approach. Using hidden representations to overcome these limitations appears more like an ad hoc solution to issues of computational overhead and broader applicability rather than a well-designed solution. A potential drawback of this version is the potential degradation of model performance (e.g., accuracy), as it resembles Dropout (or Gaussian Dropout [8]), which is known to degrade model performance [4].

* the following example illustrates the **potential ineffectiveness of the proposed approach**: consider the setup in Figure 1(b) and a point, such as (2, -4), which lies deep within the predicted class (class 0) but should have higher predicted uncertainty, as it is not near the center of the red class. Adding Gaussian noise to this point won’t change the predicted class, so the recalibrated predictions would remain highly miscalibrated. A similar issue could occur when the predicted softmax vector changes under different perturbations, but the argmax class remains nearly the same, resulting in a degenerate distribution and miscalibrated predictions. By contrast, ensembling approaches rarely exhibit this behavior.

[1] Mi, Lu, et al. "Training-free uncertainty estimation for dense regression: Sensitivity as a surrogate." AAAI 2022.

[2] Durasov, Nikita, et al. "Zigzag: Universal sampling-free uncertainty estimation through two-step inference." TMLR 2024.

[3] Conde, Pedro, et al. "Approaching test time augmentation in the context of uncertainty calibration for deep neural networks." arXiv 2023.

[4] Ashukha, Arsenii, et al. "Pitfalls of in-domain uncertainty estimation and ensembling in deep learning." ICLR 2020.

[5] Laurent, Olivier, et al. "Packed-ensembles for efficient uncertainty estimation." ICLR 2023.

[6] Antorán, Javier, et al. "Depth uncertainty in neural networks." NeurIPS 2020.

[7] Ramé, Alexandre, et al. "Mixmo: Mixing multiple inputs for multiple outputs via deep subnetworks." ICCV 2021.

[8] Gal, Yarin, and Zoubin Ghahramani. "Dropout as a bayesian approximation: Representing model uncertainty in deep learning." ICML 2016.

**Questions:**

As it was mentioned in the Weaknesses section, the paper:

1) misses a significant body of important related work and lacks novelty (which I find to be a core issue), as sensitivity-based approaches have been around for a long time.

2) lacks thorough experimental validation to support the idea that the method is indeed extendable and effective.

3) the approach has important theoretical issues that were not addressed in the method discussion.

To name a few questions:

1. How does the proposed method distinguish itself from prior perturbation-based uncertainty methods? Could its unique contributions be clarified?

2. Why were ensembling methods not included as baselines? Would a comparison, especially with efficient ensembling variants, provide a fuller performance picture?

3. How generalizable is the method beyond image classification? Could adaptations for language models or GNNs improve its applicability?

4. Have the theoretical limitations of the method been considered, and could further analysis address potential weaknesses in the approach’s foundations?

---

### Note · Authors · 2024-11-13

I have read and agree with the venue's withdrawal policy on behalf of myself and my co-authors.